# IL-10 Could Play a Role in the Interrelation between Diabetes Mellitus and Osteoarthritis

**DOI:** 10.3390/ijms20030768

**Published:** 2019-02-12

**Authors:** Sandeep Silawal, Maximilian Willauschus, Gundula Schulze-Tanzil, Clemens Gögele, Markus Geßlein, Silke Schwarz

**Affiliations:** 1Institute of Anatomy, Paracelsus Medical University, Nuremberg and Salzburg, Prof. Ernst Nathan Str. 1, 90419 Nuremberg, Germany; sandeep.silawal@pmu.ac.at (S.S.); m.willauschus@stud.pmu.ac.at (M.W.); clemens.goegele@pmu.ac.at (C.G.); silke.schwarz@pmu.ac.at (S.S.); 2Department of Biosciences, Paris Lodron University Salzburg, Hellbrunnerstr. 34, 5020 Salzburg, Austria; 3Department of Orthopedics and Trauma Surgery, Nuremberg General Hospital, Paracelsus Medical University, Nueremberg. Breslauer Strasse 201, 90471 Nuremberg, Germany; markus.gesslein@klinikum-nuernberg.de

**Keywords:** osteoarthritis, diabetes mellitus, hyperglycemia, hyperinsulinemia, IL-10, human articular chondrocytes

## Abstract

The association between osteoarthritis (OA), obesity and metabolic syndrome suggests an interrelation between OA and diabetes mellitus (DM). Little is known about the role of anti-inflammatory cytokine interleukin (IL)-10 in the interrelation between OA and DM. Hence, the effects of IL-10 under hyperglycemia (HG) and hyperinsulinemia (HI) in human articular chondrocytes (hAC) and chondrosarcoma cell line Okayama University Medical School (OUMS)-27 were examined. HAC and OUMS-27, cultured in normoglycemic (NG) and HG conditions were stimulated with insulin and/or IL-10. Cell survival, metabolic activity, proliferation and extracellular matrix (ECM) synthesis were immunocytochemically examined. No significant differences in vitality of hAC neither in pure NG (NG_w/o_) nor HG (HG_w/o_) conditions were found. Applying HI and/or IL-10 in both conditions reduced significantly the vitality of hAC but not of OUMS-27. HG impaired significantly hAC metabolism. When combined with HI + IL-10 or IL-10 alone it decreased also significantly hAC proliferation compared to NG_w/o_. In OUMS-27 it induced only a trend of impaired proliferation compared to NG_w/o_. hAC but not OUMS-27 reduced significantly their collagen type (col) I, SOX9 and proteoglycan (PG) synthesis in HG combined with HI +/− IL-10 compared to NG_w/o_. IL-10 could not moderate HI and HG effects. In contrast to hAC OUMS-27 showed limited sensitivity as DM model.

## 1. Introduction

Osteoarthritis (OA) is the most common degenerative joint disorder worldwide [1,2]. It is characterized by a multifactorial pathogenesis, where age, previous joint injuries, family history of OA corresponding to the genetic status, obesity as well as metabolic syndrome are considered as main risk factors [3] (Figure 1). Progressive degeneration of the articular cartilage in OA culminates in a complete destruction of the joint structure and a narrowing of the joint space and ultimately leads to pain and restricted mobility [4,5,6,7,8]. In addition to disturbing chondral tissue homeostasis, this heterogeneous and multifactorial “whole-joint disease” affects the integrity of all the tissues of the joint. In the course of OA pathological changes of all joint-related tissues lead to cellular and structural osseous alterations, synovitis, changes in the composition of the synovial fluid, degeneration of the ligaments in the knee and of the menisci [1,2,3,7,9,10]. The destructive events affecting joint tissues are primarily mediated by the presence of the pro-inflammatory cytokines tumor necrosis factor (TNF)α, interleukin (IL)-1β and probably also IL-18 which are known to suppress cartilage-specific extracellular matrix (ECM) synthesis and promote its degradation [8,10,11,12,13,14]. A physiological serum IL-10 concentration of 12.02 ± 7.23 pg/mL is known [15]. An increased release of IL-10 in conjunction with TNFα and IL-6 has been reported in OA patients, which may be considered as a compensatory reaction of the body to reduce inflammatory flare ups and in the synovial fluid of patients undergoing anterior cruciate ligament reconstruction [12,16,17,18,19]. Low IL-10 but high pro-inflammatory cytokine IL-6 levels were associated with higher risks for lumbar OA in postmenopausal woman [20]. IL-10 provides an inhibitory effect on IL-1β and TNFα expression, on synthesis of matrix metalloproteinases (MMPs) as well as on secretion of prostaglandin E_2_ (PGE_2_) [12,14,17] which may result in protecting articular cartilage from the degenerative processes [21,22]. It stabilized the specific phenotype of chondrocytes, protects them from cell death and ECM degradation in response to mechanical injury [23,24]. However, so far, little information about the role of IL-10 and its mechanism of action in healthy and in OA cartilage under the conditions of hyperglycemia (HG) is available [22]. 

An impaired release of anti-inflammatory and regulatory cytokines such as IL-10 also plays an important role in the worldwide epidemic obesity, a main risk factor for development and progression of OA [17,22,25,26,27]. It has long been observed that obese patients with a body mass index (BMI) more than 30 kg/m^2^ are almost seven times more likely to develop OA in knee joints than normal-weight subjects [28]. Today the direct relationship between obesity and OA is known to be not only a consequence of constant overload of weight-bearing hip and knee joints but mainly of persisting systemic low-grade inflammatory processes affecting many organs in addition to the musculoskeletal system [17,22,25]. The abnormal activation of pro-inflammatory pathways in activated white adipose tissue of obese subjects contributes to the synthesis of pro-inflammatory cytokines, such as TNFα, IL-1, IL-6, IL-8 and IL-18 promoting synovial inflammation and the release of cartilage degrading enzymes [17,27]. It is assumed that TNFα and IL-6 are to varying degrees the driving force behind insulin resistance and especially IL-6 can be regarded as a marker for the metabolic syndrome [27].

The imbalance between excessive caloric intake and physical inactivity in obesity is highly correlated with elevated insulin levels, insulin resistance and hyperglycemia which can lead to type II diabetes mellitus (T2DM) in experimental animals as well as in human subjects [29]. Despite known modern therapeutic measures and medications, the accurate modulation of blood sugar levels in T2DM patients remains a challenge [30]. Hyperglycemic or to a lesser degree, hypoglycemic phases with glucose deprivation occur in all T2DM patients but little is known about the ability of chondrocytes to adapt to such unsteady concentrations of extracellular glucose.

An adult articular cartilage is a completely aneural, avascular and alymphatic tissue and mostly separated from the vascular spaces of subchondral bone [31]. Nutrition of the intrinsic highly glycolytic articular chondrocytes, including their supply with glucose, occurs by diffusion from the synovial fluid [30,31,32]. The synovial fluid exhibits and reflects the glucose concentrations of the plasma [33]. T2DM-typical alterations in glycemia are thought to evoke similar effects on the glucose concentration of the synovial fluid and, therefore, also affect the glucose supply of the articular chondrocytes. As glycolytic cells, articular chondrocytes depend on a constant glucose delivery to facilitate an optimal cell metabolism and maintain anabolic functions, such as the homeostasis of the cartilage ECM and neo-synthesis of cartilage-specific ECM components [30,31]. Thus, chondrocytes appear to be sensitive to shifts of the glucose content of the synovial fluid under hyperglycemic and hypoglycemic conditions, as found in T2DM, which may promote degenerative changes in cartilage and its intrinsic cells, facilitating the development and progression of OA [30,31]. hAC donors and cartilage specimen vary to a great extend in age, health and activity, hence, high variations between different hAC preparations are reflected in cell culture experiments. Furthermore, hAC can rarely be isolated in sufficient amounts to perform extensive analyses on cellular responses under defined conditions. Therefore, there is a high need for reliable and reproducible cell culture systems to avoid the problems of cell number limitations and high inter-donor variability. The chondrocyte cell line Okayama University Medical School (OUMS)-27 is derived from chondrosarcoma and expresses a differentiated chondrocytic phenotype offering the capacity to synthesise cartilage-specific ECM components such as proteoglycans (PG) and collagen (col) types II, IX and X [34,35,36]. Since OUMS-27 have been successfully used as a model to study the interrelation between chondrocytes and ECM, we examined this cell line for its suitability in OA research to investigate the interrelation of OA and T2DM.

To date, the direct interplay between T2DM and OA still remains unclear on the molecular and cellular levels. Therefore, we used a cell model to examine the effects of HG and hyperinsulinemia (HI) on hAC and OUMS-27. Additionally, since little is known about the role and mechanisms of the anti-inflammatory and chondroprotective cytokine IL-10, we considered its role as potential therapeutic starting point.

## 2. Results

### 2.1. HG and HI Influence Cell Viability and Morphology

The evaluation of all live/dead staining demonstrated that high glucose content alone had no significant influence on cell viability between 24 h and 48 h (Figure 2). 24 h and 48 h after incubation in normo- as well as hyperglycemic medium (NG_w/o_, HG_w/o_), hAC (Figure 2) and OUMS-27 (Figure 3) showed a cell survival rate of more than 94%. 48 h after inducing HI by an insulin concentration of 10 µg mL^−1^ under HG conditions, the hAC exhibited a significantly impaired survival rate compared to NG_w/o_ and HG_w/o_ (Figure 2).

Inducing HI combined with IL-10 in NG led to a significant reduction of hAC cell viability within the first 24 h. After 48 h, the same effect was seen in NG, not only when HI was combined with IL-10, but also induced by IL-10 alone. Interestingly, the survival rate of hAC was below 75% in the presence of insulin and IL-10 in NG and HG after 48 h. Also the combination of hyperglycemia and hyperinsulinemia impaired the survival of hAC significantly at 48 h. Adding IL-10 alone under hyperglycemic conditions supported cell survival after 48 h at a rate of 91.2 ± 5.71% surviving cells (Figure 2) which was significantly higher than in normoglycemic conditions.

OUMS-27 cells (Figure 3) appeared to be less susceptible to HG, HI and IL-10 stimulation. 24 h and 48 h after stimulation there were no significant changes in OUMS-27 cell viability displaying a survival rate between 89.8% and 98.03% after 48 h incubation with all respective stimulation media (Figure 3). During the stimulation experiments hAC exhibited a fibroblast-like phenotype in monolayer culture under NG_w/o_ or HG_w/o_ conditions (Figure 2). 

### 2.2. Glucose Content and Insulin Stimulation Impairs Metabolic Activity and Cell Proliferation

hAC cultured in NG conditions were not as susceptible to treatment with insulin and/or IL-10 as hAC in HG conditions. The results clearly showed, that HG significantly reduced the metabolic activity of hAC in all stimulation groups (Figure 4A). Furthermore, compared with the untreated NG_w/o_ control group, the metabolic activity of all HG stimulation groups was significantly impaired. The metabolic activity of hAC cultured at NG exposed to HI alone was significantly higher compared with the HI + IL-10 treatment and the latter was significantly higher than in hAC prone to the treatment with IL-10 alone.

Metabolism of OUMS-27 cells (Figure 4B) proved to be less influenced by glucose content compared to the hAC and no significant differences were found in all stimulation groups compared to the control group (NG_w/o_, HG_w/o_).

Proliferation analyses showed no significant differences in hAC under NG compared to HG conditions. Treatment with insulin and IL-10 or IL-10 alone significantly reduced proliferation of hAC under HG culture conditions compared to untreated hAC in NG (NG_w/o_) (Figure 4C). 

In line with the results of metabolic activity, no significant differences between NG and HG culture conditions were found in the proliferation analysis of OUMS-27 (Figure 4D), even after elevating the insulin levels and/or treatment with IL-10 of OUMS-27. But the results showed the trend, that high extracellular glucose concentrations evoked a clearly reduced proliferation of OUMS-27 in each of the stimulation groups compared to the respective NG group (Figure 4D). 

### 2.3. Impaired Cartilage ECM Production under Hyperglycemic and Hyperinsulinemic Conditions

The observation of col II expression in hAC showed no significant differences except for the addition of IL-10 to HI which reduced significantly the col II synthesis in NG conditions, when compared to NG_w/o_ (Figure 5A,B). 

Furthermore, high extracellular glucose levels induced a significant reduction of col I synthesis compared to NG_w/o_, when combined with HI with or without IL-10 (Figure 5C). Within the NG conditions, only IL-10 induced a suppression of the synthesis of the dedifferentiation marker col I compared to the unstimulated control group. 

The synthesis of PG (Figure 6A,B) was significantly reduced by HI+IL-10 and IL-10 alone stimulations in hAC, regardless of the applied glycemic culture conditions. HI suppressed PG only under HG conditions in hAC. HI induced a significantly higher expression of the chondrogenic transcription factor SOX9 under NG compared with HG conditions. Nevertheless, the combination of HG conditions with HI + IL-10 or IL-10 alone led to a significantly impaired SOX9 expression compared to the NG_w/o_ (Figure 6C). 

Co-stimulation of the OUMS-27 cells under HG conditions with HI+IL-10 significantly suppressed the synthesis of the cartilage-specific marker col II in comparison to NG_w/o_ (Figure 7). The increase in insulin level under NG led to a significant reduction in col I synthesis of OUMS-27 compared to the unstimulated control (Figure 7). In contrast to hAC, the OUMS-27 cells showed no significant differences in the reaction profile in PG and SOX9 protein expression.

## 3. Discussion

To investigate the interrelation between T2DM and OA on a cellular level we used an in vitro model to characterize the responsiveness of hAC according to IL-10, insulin and high extracellular glucose levels thereby profiling the cell viability, the metabolic activity and then proliferation and synthesis of cartilage markers. The chondrosarcoma cell line OUMS-27 was tested to reveal if it represents a reliable and reproducible chondrocyte T2DM model system. These cells could circumvent donor-dependent high variability in cellular responses, effects of chondrocyte dedifferentiation during cell expansion [37] and avoid the problem of cell number limitations in an experiment. OUMS-27 cells are known to exhibit a differentiated chondrocytic phenotype and the capability to synthesise cartilage-specific ECM [34,35,36]. Although broadly used as a high grade chondrosarcoma [38,39,40,41], chondrocyte [42,43] and even OA model [44,45], only one study so far reports a direct comparison of OUMS-27 cell response with that of primary chondrocytes [46].

The immunoregulatory cytokine IL-10 is secreted by non-immune cells including articular chondrocytes [47,48] and SW1353 chondrosarcoma cells [49,50]. Articular chondrocytes and the chondrosarcoma cell line SW1353 revealed also the expression of the IL-10 receptor chains [49,51]. IL-10 exerts chondroprotective effects in cartilage e.g., by down-regulation of ECM degrading enzymes [14,47,48,52,53]. In addition, IL-10 contributed to protection of rat articular chondrocytes from apoptosis, while maintaining chondrocyte proliferation [14,47]. On the contrary, our 24 h treatment of hAC and OUMS-27 chondrocytes with IL-10 under HG conditions had a significant suppressive effect on cell survival rates of hAC which normalised after 48 h. In agreement with this, HG conditions significantly reduced the metabolic activity of the hAC but not that of OUMS-27. 

Iannone and Lapadula suggest IL-10 controls chondrocyte metabolism under physiological conditions [17]. In the present model, the metabolism of hAC was impaired by HG conditions alone as well as by HG conditions combined with HI, IL-10 or the combination of HI+IL-10. Taking into account that the regulatory capacity of IL-10 is probably concentration-dependent, another recent study showed that even overexpression of IL-10 did not moderate particular inflammation and catabolic signals such as glycosaminoglycan suppression in chondrocytes [53]. Meanwhile it is known that HG induces features of inflammation [54]. Therefore, the question arises whether other mediators are required to supplement the anabolic capacity of IL-10. Supporting this hypothesis, another study showed that IL-4 could enhance IL-10`s chondroprotective effects in form of a IL-4/IL-10 fusion protein [55].

In immune cells it is known that the immunoregulatory cytokine IL-10 can exert pleiotropic effects on cell proliferation and apoptosis, either supporting or inhibiting proliferation and apoptosis, depending on the activation state of the cells [56,57,58]. These IL-10 responses are not only cell type-dependent but probably also micromilieu-dependent and mediated either by Signal Transducer and Activator of Transcription (STAT)3 or STAT1 transcription factor activation and nuclear transition [21,58]. In OA, IL-10 exhibited an anti-proliferative influence on the chondrocytes [59]. Our study included OA-derived chondrocytes from aged donors and also confirmed this suppressive effect of IL-10 on chondrocyte cell proliferation in hAC but not in OUMS-27 cells after short term culture (48 h). 

In terms of protein expression, IL-10 is known to induce col II and aggrecan synthesis in vitro, as deduced from experiments with healthy and OA articular cartilage [14]. IL-10 has been shown to directly promote the synthesis of PG in cartilage explants [17]. Nevertheless, in our experiments, the treatment of cultured hAC with IL-10 led to a significant decrease in the non-specific and dedifferentiation associated col I (only at NG), cartilage PG (under both, NG and HG conditions) as well as the chondrogenic transcription factor SOX9 (only under HG condition) synthesis after 48 h compared to NG_w/o_. Hence, the data shows that inducing a continuous latent inflammation by HG [54] might interfere with some anabolic IL-10 actions and explain the impaired expression of chondrogenic markers observed under HG conditions. Despite hAC had to be expanded until passages 3 and 4 to gain sufficient cells to perform all experiments, they still expressed the above mentioned cartilage markers, thereby excluding their irreversible dedifferentiation [37]. However, OUMS-27 did not show any significant effect of the IL-10 treatment, suggesting their limited responsiveness in comparison to primary chondrocytes. Accordingly, Radons et al. reported previously that despite the IL-10 receptor gene was expressed in another chondrosarcoma cell line (SW1353) IL-10 did not compensate catabolic effects mediated by IL-1 and did not show activation of the major transcription factor STAT3 associated with the IL-10 pathway [49].

In chondrocytes, glucose plays a crucial role as a nutritional factor and in energy metabolism. Although glucose serves as a structural precursor for the ECM synthesis [60], high levels of extracellular glucose affect the vitality and chondrogenic potential of chondrogenic cells [22]. Extracellular high glucose levels are known to execute harmful effects due to immoderate intracellular glucose accumulation [30]. This excessive glucose conglomeration can saturate the glycolytic pathway, triggering other secondary pathways participating in the cell’s glucose metabolism. These processes lead to increased production of advanced glycation end products (AGEs), which in turn are responsible for an increased intracellular oxidative stress accompanied by a heightened Reactive oxygen species (ROS) production. Besides mitochondrial dysfunctions due to excessive glucose accumulation, AGEs and ROS are also known to derogate cartilage homeostasis and be involved in the progression of OA [22,61]. 

In a study presented by Heywood et al. [62] bovine articular chondrocytes were expanded for more than 30 days under NG as well as HG conditions. Under NG conditions chondrocytes proliferated more slowly compared to HG conditions [62]. Monitoring a smaller time course of 48 h and using human cells, the present study could not confirm this observation of Heywood et al. and in contrast to Heywood et al., we found a significant suppressive effect of HG on chondrocyte proliferation when combined with IL-10 alone or with HI + IL-10. This influence on proliferation was in agreement with a significantly diminished metabolic activity under HG compared to NG conditions already within the short observation period of 48 h. 

Furthermore, Heywood et al. demonstrated that the glucose content influenced not only chondrocyte proliferation but also the gene expression of the differentiation markers col II and SOX9 after 4 population doublings. Genes of both chondrocyte phenotype markers were higher expressed under NG conditions than under HG. Focusing on 48 h and the protein expression of these markers the present study revealed no significant difference between HG and NG conditions except for a significant higher SOX9 expression in NG in the presence of high insulin levels which could corroborate the observations of Heywood et al [62]. 

Insulin, produced by the β-cells of pancreas and reaching the joint via the capillaries of the synovial membrane, regulates carbohydrate and fat metabolism and also stimulates anabolic responses in cartilage [36,63]. Under NG conditions the treatment of healthy with different concentrations of insulin (1 nM and 10 nM insulin) for 48 h increased 2-Deoxy-d-glucose uptake as well as up-regulation of GLUT-1 in a concentration-dependent manner relative to untreated cells [64]. The chondrocytes stimulated with anabolic factors such as insulin or IGF-1, may redirect accumulated glucose into metabolic and structural factors [60].

Today, it is known that insulin acts through insulin receptors (InsR) to play a pivotal role in chondrocyte differentiation and possesses a physiological role in regular growth of bone length [65]. As demonstrated by gene and protein expression analyses for col X and PG, induction of HI at low doses of 10–50 nM in metatarsal explants stimulates hypertrophic differentiation [65]. In previous studies, insulin treatment facilitated some anabolic responses such as upregulated col II expression in chondrocytes [63,64,66], which are known to be physiologically regulated by activation of the InsR. Nevertheless, the present study using HI could not confirm a significant stimulatory effect on col II neither in hAC nor in the OUMS-27 cell line after short-term stimulation (48 h) with insulin.

In our study, diabetic conditions characterized by elevated glucose as well as insulin levels, evoked a clearly diminished PG and col I synthesis in hAC. In addition, SOX9 expression was significantly lower in HG compared to NG conditions. These observations show a suppressive effect on chondrocyte synthetic capacity as expected in T2DM, here represented as HG in the presence of HI.

These results are in agreement with Ribeiro et al., who stimulated human cartilage explants under HG conditions with insulin (100 + 1000 nM), characteristic for insulin resistance and T2DM, resulting in a loss of PG and a concomitant increase in MMP-13 expression [67]. MMP-13 plays a key role in ECM degradation and catabolic processes specifically by cleaving col II fibres and thus promoting cartilage degradation under OA. Other researchers even reported that insulin treatment at lower concentrations than in the present study (1–10 ng/mL versus 10 µg/mL) promoted col II but not aggrecan gene expression in primary hAC [64].

Accordingly, cell culture experiments based on rat- or human-derived (OUMS-27) chondrosarcoma cells with HI (10 µg/mL) [36] or lower levels (1 µg/mL insulin) [68] demonstrated the dependence of chondrosarcoma cell growth, PG and collagen synthesis on insulin [36,68]. Stimulation of chondrosarcoma derived chondrocytes with physiological insulin concentrations evoked an increased PG, col II and hyaluronan synthesis [68]. Considering that immortalized chondrosarcoma cell lines might display a response differing from hAC as also suggested by Otsu et al. [68], the effect of insulin is surely concentration-dependent and probably also slightly influenced by the source of insulin (human-/porcine-/bovine derived). 

## 4. Materials and Methods

### 4.1. Isolation of Human Primary Articular Chondrocytes (hAC)

Human articular cartilage for enzymatic isolation of human articular chondrocytes (hAC) was obtained from OA patients undergoing total hip and knee replacement surgeries or surgical interventions due to cartilage defects in the knee joint (donor *n* = 10, gender ratio male:female = 8:2; mean age: 68.8 ± 20.52 years). Cartilage harvesting was approved by the Charité–Universitätsmedizin Berlin, Ethical Committee (Dep. of Orthopaedic, Trauma and Reconstructive Surgery, Charité, CBF EA4/063/06 and Klinikum Nuremberg Medical School gGmbH [Bavarian Medical Association, No. 17029], approval date: 8 August 2017) and conducted in compliance with university ethics guidelines as well as German federal and state law. Immediately after explantation, human donor cartilage was rinsed in sterile phosphate buffered saline (PBS [Biochrom AG/Merck, Berlin, Germany] containing 1% gentamycin [Biochrom AG/Merck] and 1% penicillin/streptomycin [Biochrom AG/Merck]. Prior to digestion, cartilage specimen was minced and pre-digested in 2% pronase digestion medium (Dulbecco’s modified Eagle’s medium [DMEM]/Ham’s F12 w/o foetal bovine serum (FBS) [Biochrom AG/Merck] supplemented with 20 mg/mL pronase [Serva, Heidelberg, Germany]) for 1 h at 37 °C. Subsequently, the tissue specimen was washed twice with sterile PBS and centrifuged at 300× *g*. After discarding the supernatant, the pre-treated tissue specimen was digested overnight at 37 °C with collagenase solution (0.1% collagenase NB5 [Serva] diluted in culture medium DMEM/Ham’s F12 with 0.5% FBS [Biochrom AG/Merck]). After using 100 µm pore sized cell strainer (Sigma-Aldrich, Saint Louis, MO, USA), the cell suspension was centrifuged at 300× *g*. The total cell number and vitality was determined by trypan blue exclusion test.

In comparison to primary human articular chondrocytes OUMS-27 chondrosarcoma cells were applied (IFO 50488, JCRB Cell Bank, Ibaraki, Osaka, Japan).

### 4.2. Expansion Culture of hAC and OUMS-27

Immediately after isolation, hAC were seeded in cell culture flasks (CellPlus, Sarstedt AG, Nümbrecht, Germany) and grown in monolayer culture with basal standard culture medium (DMEM: HAM’s-F12 1:1 supplemented with 10% FBS, 1% amphotericin B, 1% penicillin/streptomycin, 14.2 mM ascorbic acid, MEM non-essential amino acids [Sigma-Aldrich, Saint Louis, MO, USA]). When reaching 80–90% confluence, the cells were detached by trypsin/EDTA solution [Biochrom AG/Merck] (0.05%/0.02%), counted and cryopreserved.

For the stimulation experiments hAC and OUMS-27 were thawed, seeded at an initial cell density of 1.0 × 10^4^ cells cm^−2^ into 175 cm^2^ cell culture flasks and grown in monolayer culture until 80% confluence. For all experiments hAC in passage 3 to 4 were used.

### 4.3. Stimulation Culture

Round cover slips (d = 12 mm) were washed with 70% ethanol and autoclaved. After sterilization, cover slips were rinsed twice in sterile PBS and coated with poly-L-lysin (1:100 in 1× PBS) for 20 min at room temperature (RT). Subsequently, coverslips were rinsed with sterile H_2_O_deion_ and dried. Prior to cell seeding, cover slips were washed twice in sterile PBS at RT and transferred individually into 24-well plates. 

Cells were seeded onto the cover slips (initial cell number of 1 × 10^4^ cm^−2^) and onto 96-well-plates (initial cell number of 7.5 × 10^3^ cells/well). Prior to stimulation with insulin from bovine pancreas (Sigma-Aldrich) 10 µg mL^−1^ or recombinant human IL-10 (PeproTech, Hamburg, Germany) 10 ng mL^−1^ or co-stimulation (insulin [HI] + IL-10), cells were allowed to attach in standard culture medium for 24 h at 37 °C and 5% CO_2_. After 24 h of incubation, cells were serum-starved in normoglycemic (DMEM, FG 0415, Biochrom AG) and hyperglycemic (DMEM, FG 0445, Biochrom AG) medium 1.0 g L^−1^ glucose and 4.5 g L^−1^ glucose respectively containing 1% FBS for 2 h at 37 °C and 5% CO_2_. The stimulation was performed with the respective NG and HG culture media containing 1% FBS for 24 and 48 h at 37 °C and 5% CO_2_ as well as the respective treatments (stimulation groups see Table 1).

### 4.4. Cell Survival

To examine the vitality of the unstimulated and stimulated cells after 24 and 48 h in stimulation culture, live/dead staining using 1 µL propidium iodide (PI, 1% stock solution) (ThermoFisher Scientific, Darmstadt, Germany) and 5 µL fluorescein diacetate (FDA, stock solution: 3 mg mL^−1^ in acetone) (Sigma-Aldrich, USA) in 1 mL 1× PBS was performed. For performing the live/dead staining, the seeded cover slips were removed from medium, 50 µL of stain solution was applied and transferred to a microscopic cover slide. After a 5 min incubation period at RT, the fluorescence of live and dead cells was monitored using a Leica TC SPEII confocal laser scanning microscope (Leica, Wetzlar, Germany).

### 4.5. Quantitative Assays for DNA Quantification and Assessment of Cell Metabolism

The differences in DNA quantification was assessed after 48 h under different stimulating conditions applying the CyQUANT^®^ NF Cell Proliferation Assay Kit. Changes in cell metabolism were examined by using CellTiter-Blue^®^ Cell Viability Assay. Both assays were used according to the manufacturer’s instructions. 

Initially, 7500 cells/well were seeded onto a 96-well plate. For each stimulation 6 wells were seeded for cell viability assay as well as cell proliferation assay, so each stimulation was measured as sextet.

Cells were allowed to adhere in respective culture medium for 24 h at 37 °C, 5% CO_2_ and under humidified conditions. After 24 h incubation, medium was removed completely by tapping and cells were serum-starved in respective culture medium containing 1% FBS for 2 h at 37 °C and 5% CO_2_. Subsequently, the cells were washed two times in sterile 1× PBS. To ensure that all surplus PBS was removed, well plates were tapped on sterile, extremely absorbent paper. Subsequently, cells were stimulated with the respective NG and HG stimulation media shown in Table 1 for 48 h at standard culture conditions.

#### 4.5.1. The CellTiter-Blue^®^ Cell Viability Assay

For the estimation of differences in metabolic capacity of cells after stimulation CellTiter-Blue^®^ Cell Viability Assay was applied after 8 h of stimulation. 100 µL of the respective stimulation medium per well was mixed with 25 µL of Alamar blue solution and cells incubated for further 6 h. Blue non-fluorescent resazurin contained in the incubation solution added to the growth medium penetrates the cells and is reduced by several intracellular (mitochondrial, cytosolic and microsomal) redox enzymes into red and highly fluorescent resorufin depending on their cellular activity rate. The fluorescence of each sample was measured in sextet at 560 excitation/590 emission nm in a fluorometric plate reader (Infinite M200 Pro, Tecan, Groedig, Austria).

#### 4.5.2. DNA Quantification of Adherent Cells

By CyQUANT^®^ NF Cell Proliferation Assay the influence of respective treatment on cell division was examined after 48 h of stimulation. The standard curve was generated by serial dilution of calf thymus DNA stock solution (1 mg mL^−1^) with TE-buffer (TRIS EDTA buffer, 10 mM TRIS [pH 8.0], 1 mM EDTA in H_2_O_deion_.). After 48 h of stimulation, medium was completely removed and cells were washed carefully once with 1× Hank’s Balanced Salt Solution (HBSS, Gibco, Paisley, UK). Subsequently, HBSS was thoroughly removed and 50 µL of the dye solution (1× HBSS + dye binding solution 1:500) was applied to each seeded well. For the standard curve, 25 µL of the serial calf thymus DNA dilutions was mixed with 25 µL of CyQuant dye solution (1× HBSS + dye binding solution 1:250, ThermoFisher Scientific). 

Subsequently, plates were covered to protect from light and incubated at 37 °C for 60 min. The fluorescence of each well was measured in triplicate at 485 excitation/530 emission nm in a fluorometric plate reader.

### 4.6. Immunocytochemical Detection of Cartilage-Specific Markers

For immunocytochemical detection of cartilage-specific ECM markers, after 24 and 48 h of stimulation culture on cover slips, cells were rinsed twice for each 5 min at RT in sterile 1× PBS and fixed in 4% paraformaldehyde solution (PFA) for 20 min at RT. After fixation cover slides were rinsed in 1× PBS again and could be stored at 4 °C until further use. Prior to immunocytochemical staining, cover slides were washed three times for 5 min at RT with 1× TRIS buffered saline (TBS). For permeabilization and blocking, cells were incubated in TBST blocking solution (1× TBS with 5% donkey serum and 0.1% Triton X100, TBST) for 1 h at RT. Subsequently, cells were washed twice with TBS and primary antibody solution was applied (antibodies in TBST blocking solution, dilution factor see Table 2) for 16 h at 4 °C in a humified chamber. Double staining of col I and col II as well as human adult cartilage PG (recognizing the peptides substituted with keratan sulphate side chains) with SOX9 (antibody specification and applied dilutions see Table 2) was performed.

### 4.7. Statistical Analysis

Statistical analysis was conducted using GraphPad Prism 7 (GraphPad Software, Inc., San Diego, CA, USA) whereby normoglycemic cultured cells without any stimulation “NG_w/o_” were used as the control group for all performed statistical tests. Before testing normal distribution of the results, the ROUT method of identifying outliers was applied. The normal distribution of the results was analysed using the Shapiro Wilk test as well as the D’Agostino and Pearson normality test. When the normality test failed, the Kruskal–Wallis one-way analysis of variance on ranks followed by Dunn’s post hoc multiple comparisons was performed to evaluate the significance of the performed analyses of cell viability, proliferation, relative metabolic activity and ECM neo-synthesis between stimulation groups. In the case of a normal distribution, the ANOVA one-way analysis was used for the statistical evaluation of the data (level of significance α = 0.05). The relative percentage of cell vitality, proliferation and relative metabolic activity are reported as mean values ± standard deviation (SD).

## 5. Conclusions

In conclusion, our results show that high levels of extracellular glucose impair not only the vitality but also the metabolism and proliferation of hAC. Exposure to HG concentrations accompanied with HI and IL-10 impaired the synthesis of cartilage-specific ECM components PG and SOX9. The administered supraphysiological concentration of IL-10 (10 ng/mL) was not sufficient to restore cell proliferation and metabolic activity of hAC and OUMS-27 under HG. Hence, IL-10 could not exert chondroprotective properties under the tested conditions. As HG obviously diminished the chondroprotective abilities of IL-10, other concentrations should be tested in future e.g. by overexpression strategies and combined with other anti-inflammatory mediators to achieve a therapeutic chondroprotective effect on hAC. Since in this study only short-term effects of HG, HI and IL-10 could be monitored in monolayer culture, ongoing studies will assess their long term effects by using a suitable three dimensional chondrocyte culture model.

The chondrosarcoma cell line OUMS-27 was established as a useful tool for investigating therapeutic approaches for chondrosarcomas and is broadly used as in vitro OA model. The present study reflected only a few of the effects observed in primary hAC. Hence, it is important to take into account distinct differences regarding the cell metabolism, survival capacity and protein regulation of OUMS-27 cells in comparison to primary hAC. Based on our results we cannot recommend this cell line as T2DM model for OA research.

## Figures and Tables

**Figure 1 ijms-20-00768-f001:**
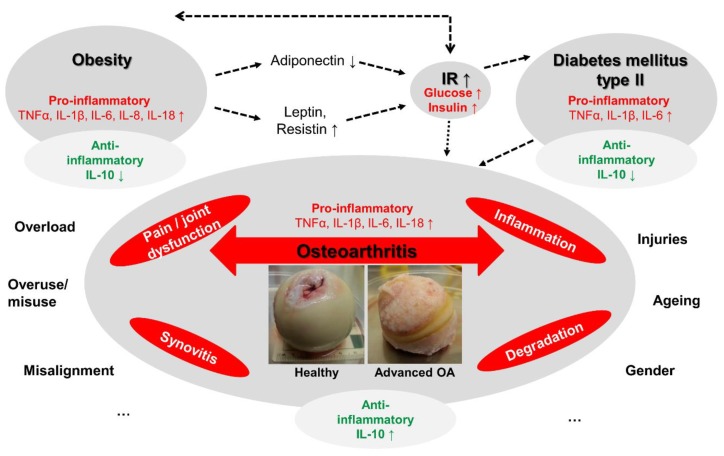
Scheme of the putative interrelation between osteoarthritis, obesity and diabetes mellitus type 2. IL: interleukin, IR: insulin resistance, TNFα: tumour necrosis factor alpha. ↓: decrease, ↑: increase. Green: anti-inflammatory, red: pro-inflammatory. Dotted arrows: induction by. …: should suggest that there exist a couple of other pre-disposing factors contributing to OA pathogenesis.

**Figure 2 ijms-20-00768-f002:**
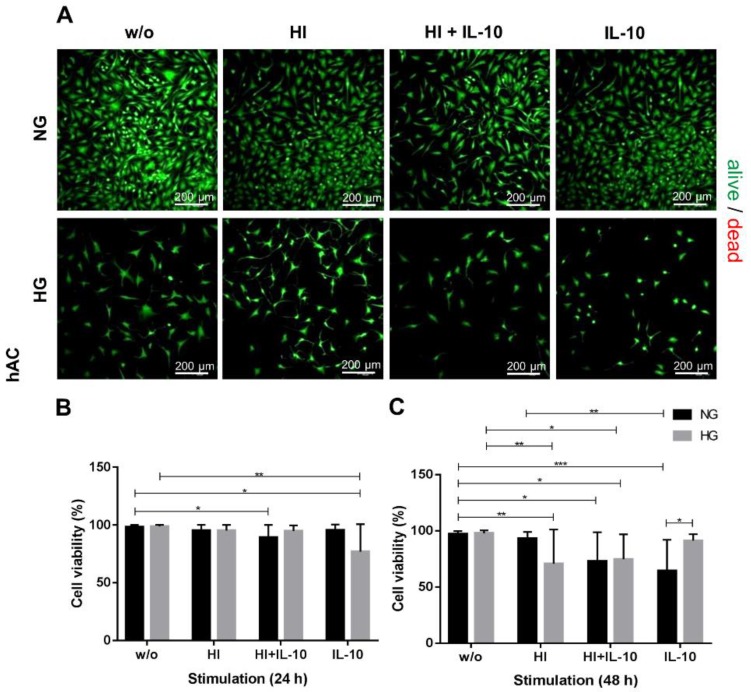
Primary human articular chondrocyte (hAC) survival in response to hyperglycemia (HG), hyperinsulinemia (HI) and IL-10 stimulation. Live/dead staining 24 h (**A**) and cell viability of hAC calculated after 24 h (**B**) and 48 h (**C**) culture at HG (light grey) or normoglycemia (NG, dark grey) conditions as well as with respective stimulation. Cells alive: green, dead cells: red, w/o: without any stimulation; NG: 1.0 g/L glucose; HG: 4.5 g/L glucose; 10 µg/mL; IL-10: interleukin-10, 10 ng/mL). * *p* < 0.05, ** *p* < 0.01, *** *p* < 0.001. (**A**): scale bars: 200 µm. *n* = 8: independent experiments performed with hAC of different donors.

**Figure 3 ijms-20-00768-f003:**
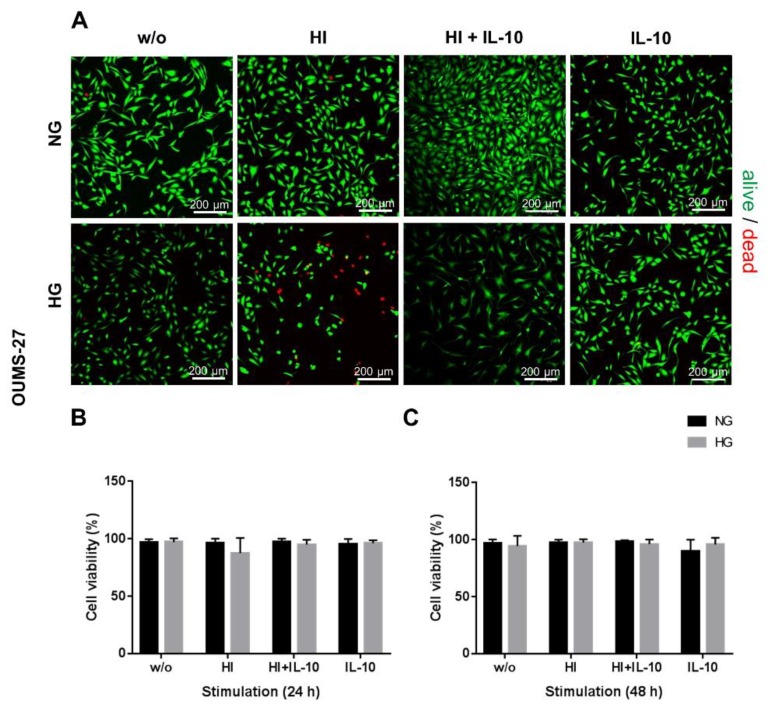
OUMS-27 chondrocyte survival in response to hyperglycemia (HG), hyperinsulinemia (HI) and IL-10 stimulation. Live/dead staining (**A**) and cell viability of OUMS-27 after 24 (**B**) and 48 h (**C**) culture at HG (light grey) or NG (dark grey) conditions as well as induction of HI and/or stimulation with IL-10. w/o: without any stimulation; NG: 1.0 g/L glucose; HG: 4.5 g/L glucose; HI: 10 µg/mL; IL-10: interleukin-10, 10 ng/mL). (**A**): scale bars: 200 µm. *n* = 4: independent experiments.

**Figure 4 ijms-20-00768-f004:**
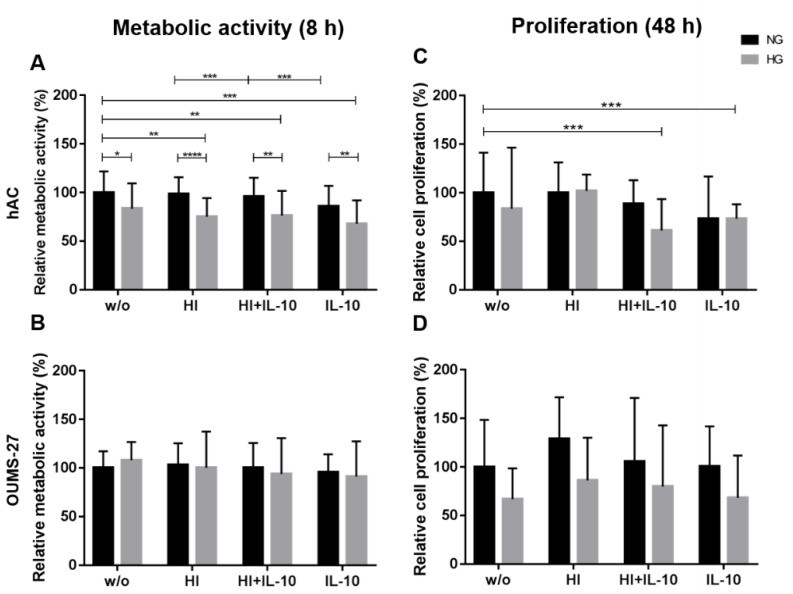
Relative metabolic activity (**A** + **B**) and cell proliferation (**C** + **D**) of hAC (**A** + **C**) and OUMS-27 cells (**B** + **D**) respectively under hyperglycemia (HG, light grey) or normoglycemia (NG, dark grey) conditions alone or after stimulation with hyperinsulinemia (HI) with or without IL-10 or IL-10 alone. w/o: without any stimulation; NG: 1.0 g/L glucose; HG: 4.5 g/L glucose; HI: 10 µg/mL; IL-10: interleukin-10, 10 ng/mL). The metabolic activity as well as proliferation of the cells under NG_w/o_ has been normalised to 100%. * *p* < 0.05, ** *p* < 0.01, *** *p* < 0.001, **** *p* < 0.0001. *n* = 10 (**A**), *n* = 4 (**B** + **D**), *n* = 8 (**C**): independent experiments performed with hAC of different donors and OUMS-27.

**Figure 5 ijms-20-00768-f005:**
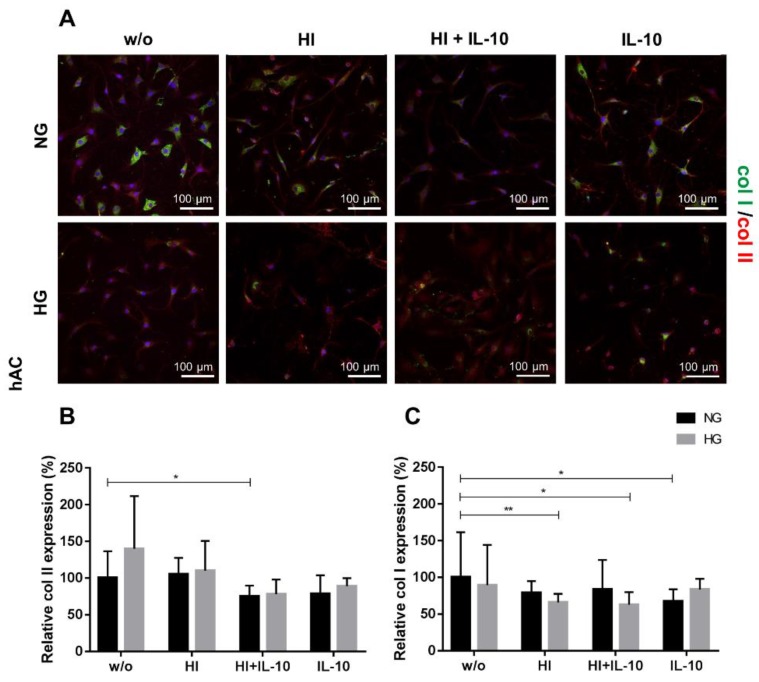
Collagen type II and I expression in response to hyperglycemia (HG), hyperinsulinemia (HI) and IL-10 stimulation in hAC. (**A**) Immunocytochemical staining for collagen type II (red) and collagen type I (green) after 48 h stimulation of hAC at HG (light grey) or NG (dark grey) conditions combined with HI and/or IL-10 stimulation or IL-10 alone. Blue: DAPI, cell nuclei; w/o: without any stimulation; NG: normoglycemic, 1.0 g/L glucose; HG: 4.5 g/L glucose; HI: 10 µg/mL; IL-10: interleukin-10, 10 ng/mL. Scale bars: 100 µm. (**B**) Evaluation of relative protein expression by monitoring the fluorescence intensity of collagen type II (**B**) and type I (**C**) synthesis after 48 h of respective stimulation. The protein expression under NG_w/o_ has been normalised to 100%. * *p* < 0.05, ** *p* < 0.01. *n* = 5: independent experiments performed with hAC of different donors.

**Figure 6 ijms-20-00768-f006:**
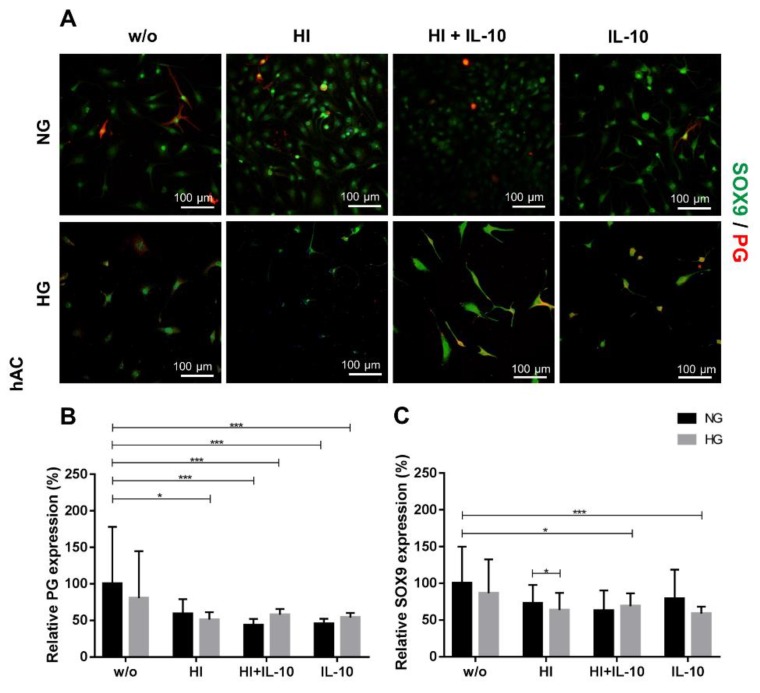
Proteoglycan (PG) and SOX9 expression in response to hyperglycemia (HG), hyperinsulinemia (HI) and IL-10 stimulation in hAC. (**A**) Immunocytochemical staining of hAC for PG (red) and SOX9 (green) after 48 h culture at HG (light grey) or NG (dark grey) conditions combined with HI and/or IL-10 stimulation or IL-10 alone. Blue: DAPI, cell nuclei; w/o: without any stimulation; NG: normoglycemic, 1.0 g/L glucose; HG: 4.5 g/L glucose; HI: 10 µg/mL; IL-10: interleukin-10, 10 ng/mL. Scale bars: 100 µm. Evaluation of relative protein expression by analysing the fluorescence intensity of PG (**B**) and SOX9 (**C**) synthesis after 48 h of respective stimulation. The protein expression under NG_w/o_ has been normalised to 100%. * *p* < 0.05, *** *p* < 0.001. *n* = 5: independent experiments performed with hAC of different donors.

**Figure 7 ijms-20-00768-f007:**
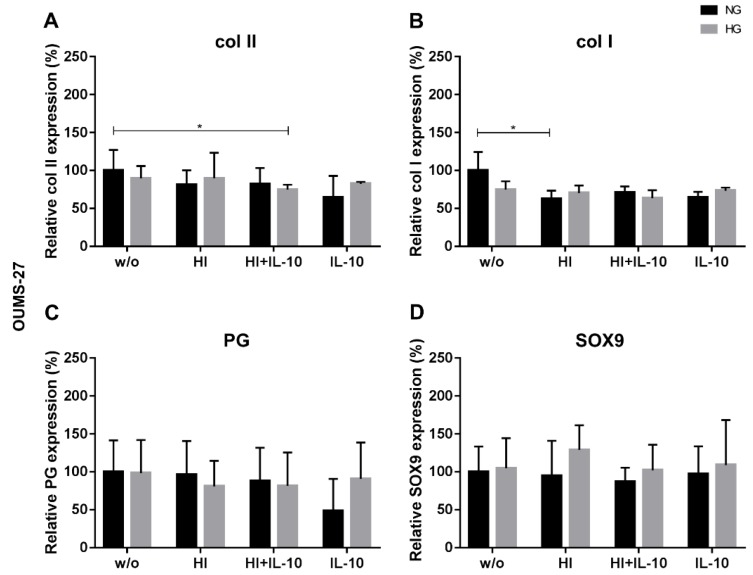
Collagen type II, I, proteoglycan (PG) and SOX9 expression in OUMS-27 in response to hyperglycemia (HG), hyperinsulinemia (HI) and IL-10 stimulation. Evaluation of relative protein expression by analysing the fluorescence intensity of collagen (col) type II (**A**), type I (**B**), PG (**C**) and Sox9 (**D**) synthesis in OUMS-27 after 48 h of respective stimulation at HG (light grey) or NG (dark grey) conditions with insulin and/or IL-10. w/o: without any stimulation; NG: normoglycemic, 1.0 g/L glucose; HG: 4.5 g/L glucose; HI: 10 µg/mL; IL-10: interleukin-10, 10 ng/mL. The protein expression under NG_w/o_ has been normalised to 100%. * *p* < 0.05. *n* = 3.

**Table 1 ijms-20-00768-t001:** Stimulation groups under normo- and hyperglycemic conditions. The normoglycemic cultured cells without any stimulation (NG_w/o_) were used in all analyses as a control group.

Without Stimulation (w/o)	HI	HI+IL-10 (Co-Stimulation)	IL-10
NG (Control Group)	NG + 10 µg mL^−1^ insulin	NG + 10 µg mL^−1^ insulin +10 ng mL^−1^ IL-10	NG + 10 ng mL^−1^ IL-10
HG	HG + 10 µg mL^−1^ insulin	HG + 10 µg mL^−1^ insulin + 10 ng mL^−1^ IL-10	HG + 10 ng mL^−1^ IL-10

**Table 2 ijms-20-00768-t002:** Applied primary and secondary antibodies and respective applied dilutions.

Target	Primary Antibody	Dilution	Secondary Antibody	Dilution
col I	goat anti human, Biozol, Eching, Germany	1:50	donkey anti goat, Alexa Fluor 555, ThermoFisher Scientific, Darmstadt, Germany	1:200
col II	rabbit anti human, Acris, Herford, Germany	1:50	donkey anti rabbit, Alexa Fluor 488, ThermoFisher Scientific, Darmstadt, Germany	1:200
PG	mouse anti human, Merck, Darmstadt, Germany	1:70	donkey anti mouse Cy3, Dianova, Hamburg, Germany	1:200
SOX9	rabbit anti human, Merck, Darmstadt, Germany	1:100	donkey anti rabbit, Alexa Fluor 488, Dianova, Hamburg, Germany	1:200

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
