# Peer review of "IL-10 Could Play a Role in the Interrelation between Diabetes Mellitus and Osteoarthritis"

_ijms, 2019, doi:10.3390/ijms20030768_

Round 1
Reviewer 1 Report
The authors studied if IL-10 plays a role in the interrelation between T2DM and OA and used primary human osteoarthritic chondrocytes (hAC) and OUMS-27 chondrosarcoma cells as tools. While the study appears interesting, to my mind some aspects would benefit from clarification:
Kunisada et al. (Int J Cancer, 1998) describe reproducible chondrosarcoma formation in mice as a “chondrocytic phenotype”, while the cells express collagen type 1 (mRNA). Other chondrosarcoma cells like RCS’, displaying a stable differentiated chondrocyte-like phenotype, do not. To my mind, OUMS-27 cells may have little in common with hACs, which is confirmed by this study (L 343) and may be more clearly addressed in L 483-488.
L 298 vs 307; “cells proliferated slower in NG” and “showed a diminished metabolic activity in HG” sounds odd to me. Please define “metabolic activity”.
L 455; is it human specific cartilage PG or human cartilage specific PG? Anyway, which proteoglycan is that? Please specify.
L 457, Tab 2; check carefully, please, as detection of col II appears artefactual as such.
L 475; what is your definition of “physiological” derived at as 10ng IL-10/ml may be very high in vivo?
L 477; “obviously” appears bold in light of the data.
Fig. 3; OUMS-27 show the typical elongated polygonal morphology, as do hACs in Fig. 2. Primary chondrocytes were used de-differentiated, in P3-4, which to my mind confines some conclusions. This should be discussed.
Fig. 5; legend (A) does not make sense and how were the Y-axes normalized? Please also indicate the number of replicates (all) the figures were derived at.
T2DM (L 222) may benefit from introduction. L 225 is a very long and hard to read sentence. Discussion is long and could be better focused. L 276-280; what is this supposed to be supportive of? Note that letter size fluctuates throughout this manuscript.
Author Response
General comment of the reviewer:
The authors studied if IL-10 plays a role in the interrelation between T2DM and OA and used primary human osteoarthritic chondrocytes (hAC) and OUMS-27 chondrosarcoma cells as tools. While the study appears interesting, to my mind some aspects would benefit from clarification:
Comment 1: Kunisada et al. (Int J Cancer, 1998) describe reproducible chondrosarcoma formation in mice as a “chondrocytic phenotype”, while the cells express collagen type 1 (mRNA). Other chondrosarcoma cells like RCS’, displaying a stable differentiated chondrocyte-like phenotype, do not. To my mind, OUMS-27 cells may have little in common with hACs, which is confirmed by this study (L 343) and may be more clearly addressed in L 483-488.
Response 1: The dilemma that the OUMS-27 cell line reflects only few of the effects observed with primary articular chondrocytes in this study is pronounced more clearly now (see changes in the conclusion: the former first sentence „and reflects a similar trend“ was omitted, Line 478-479: „Based on our results we cannot recommend this cell line as T2DM model for OA research”.
Comment 2a: L 298 vs 307; “cells proliferated slower in NG” and “showed a diminished metabolic activity in HG” sounds odd to me.
Response 2a: We state exactly now that our results concerning proliferation under NG and HG differed from the study of Heywood et al., and we clarified that this went along with a suppressive effect on metabolism (see lines 288-292).
Comment 2b: Please define “metabolic activity”.
Response 2b: It is measured as enzymatic activity and explained in lines 418-422 now „ Blue non-fluorescent resazurin contained in the incubation solution added to the growth medium penetrates the cells and is reduced by several intracellular (mitochondrial, cytosolic and microsomal) redox enzymes into red and highly fluorescent resorufin depending on their cellular activity rate.“ Thus, rasazurin detects cell metabolism by conversion into the bright red fluorescent dye resorufin within the culture medium in response to chemical reduction resulting from cellular growth and metabolism. This assay does not address any specific enzymatic activity but multiple intracellular metabolic reactions making this assay suitable as an indicator of cell viability.
Resazurin penetrates the cells and is reduced inside of viable cells by several different redox enzymes (mitochondrial, cytosolic and mictrosomal enzymes) to resorufin. Non-viable, damaged and stimulated cells often rapidly lose their metabolic capacity, are not longer able to reduce resazurin, and thus generate no or only a diminshed fluorescent signal, depending on metabolic activity of cells.
Comment 3: L 455; is it human specific cartilage PG or human cartilage specific PG? Anyway, which proteoglycan is that? Please specify.
Response 3: „human adult cartilage proteoglycan“. This is exactly the specification given in the datasheet of the antibody. Further notice declares: This antibody recognizes the short peptides substituted with keratan sulfate side chains and within the core protein of proteoglycans in articular cartilages (the information is added in line 447).
Comment 4: L 457, Tab 2; check carefully, please, as detection of col II appears artefactual as such.
Response 4: We thank the reviewer for announcing this mistake. The secondary antibody used for Col II detection was donkey anti rabbit-Alexa 488. We corrected Table 2 (Line 449) accordingly.
Comment 5: L 475; what is your definition of “physiological” derived at as 10 ng IL-10/ml may be very high in vivo?
Response 5: We thank the reviewer for adressing this error. The term „physiological“ obviously does not make sense. The physiological serum level and an incrase in response to joint pathologies in the synovial fluid is mentioned in the introduction section in line 45-46 now. In the conclusion section the sentence is corrected (line 466).
Comment 6: L 477; “obviously” appears bold in light of the data.
Response 6: We agree with the reviewer and this statement has been removed in the conclusion section.
Comment 7: Fig. 3; OUMS-27 show the typical elongated polygonal morphology, as do hACs in Fig. 2. Primary chondrocytes were used de-differentiated, in P3-4, which to my mind confines some conclusions. This should be discussed.
Response 7: Since a substantial expansion of chondrocytes was required to cover all experimental settings in the present study chondrocytes of passage (P) 3-4 had to be used. P4 can be accepted as a kind of threshold for experiments (Schulze-Tanzil, 2009). After P4 chondrocytes can undergo irreversible dedifferentiation (Schuze-Tanzil, 2009). However, in P4 chondrocytes still express cartilage markers. We discuss this matter now (lines 265-268).
Comment 8: Fig. 5; legend (A) does not make sense and how were the Y-axes normalized? Please also indicate the number of replicates (all) the figures were derived at.
Response 8: The legends have been revised. We added the number of independent experiments performed with chondrocytes derived from different donors in each figure legend. The mode of normalization is described now.
Comment 9: T2DM (L 222) may benefit from introduction. L 225 is a very long and hard to read sentence. Discussion is long and could be better focused. L 276-280; what is this supposed to be supportive of? Note that letter size fluctuates throughout this manuscript.
Response 9: L71-72: the abbreviation and typical features of T2DM have been introduced now. The mentioned sentence has been separated into two parts. This makes it easier for the reader to comprehend the text. The sentences in former lines 276-280 have been removed. The discussion has been reduced to make it more precise. The letter size has been made uniform.
Reviewer 2 Report
rows 140-145: "The initial cell...respective HG stimulation" are Materials and Methods. Delete this phrase
rows 156-157: "to examine..was applied" are Materials and Methods. Delete this phrase
row 170: writing mistake (prolferative)
rows 176-182: figure 5A and 5B indications are missing in text. Which is the result in the OUMS-27?
rows 209-213: move to the paragraph and figures (fig 7 - A and B) between lines 176-182
row 222: specify T2DM
row 228: T2DM or DMT2?
row 243: writing mistake (8 h)
row 283: specify GAGs
row 28290: specify ROS
rows 350-351: introduce BMI
row 359: specify FBS
rows 360-362: "digested overnight"...room temperature or other?
rows 363-364: "the cell suspension was centrifugated"...how many g?
Mainly: review the company names, for example: Sigma, Darmstadt, Germany not only Sigma
Author Response
Comment 1: rows 140-145: "The initial cell...respective HG stimulation" are Materials and Methods. Delete this phrase
Response 1: The phrase has been removed.
Comment 2: rows 156-157: "to examine.. was applied" are Materials and Methods. Delete this phrase
Response 2: The two sentences have been deleted.
Comment 3: row 170: writing mistake (prolferative)
Response 3: We thank the reviewer. The mistake has been corrected.
Comment 4: rows 176-182: figure 5A and 5B indications are missing in text. Which is the result in the OUMS-27?
Response 4: The missing figure indications have been introduced in the text (line 170). The OUMS27 results are described in lines 202-207.
Comment 5: rows 209-213: move to the paragraph and figures (fig 7 - A and B) between lines 176-182
Response 5: It has been checked that Fig. 7A and B are properly discribed in the result section. The mentioned phrase in the discussion has been shortened for a better match.
Comment 6: row 222: specify T2DM
Response 6: After explaining the abbreviation in line 72, only T2DM is used throughout the whole manuscript.
Comment 7: row 228: T2DM or DMT2?
Response 7: T2DM is correct. This has also been changed.
Comment 8: row 243: writing mistake (8 h)
Response 8: This has been removed.
Comment 9: row 283: specify GAGs
Response 9: Now line 243: we resigned to abbreviate it now, hence, it is fully written.
Comment 10: row 28290: specify ROS
Response 10: It has been specified now and also been included in the abbreviation list.
Comment 11: rows 350-351: introduce BMI
Response 11: We agree with the reviewer that the BMI would be a very helpful information, but unfortunately, the BMI is not known for all donors included.
Comment 12: row 359: specify FBS
Response 12: It has now been introduced in line 348 now and it is explained in the abbreviation list.
Comment 13: rows 360-362: "digested overnight"...room temperature or other?
Response 13: Line 351: 37°C: it was already mentioned, hence, no change is required.
Comment 14: rows 363-364: "the cell suspension was centrifugated"...how many g?
Response 14: The centrifugation step velocity of 300 g has been added (line 355).
Comment 15: Mainly: review the company names, for example: Sigma, Darmstadt, Germany not only Sigma.
Response 15: The changes have been made as recommended. Each company is detailed with town and country, where it is localized, when mentioned the first time.